



# Continuous methane concentration measurements at the Greenland Ice Sheet-atmosphere interface using a low-cost low-power metal oxide sensor system

Christian Juncher Jørgensen[1], Jacob Mønster[2], Karsten Fuglsang[2], Jesper Riis Christiansen[3]

[1]Department of Bioscience, Arctic Environment, Aarhus University, Roskilde, 4000, Denmark
[2]FORCE Technology, Brøndby, 2605, Denmark
[3]Department of Geoscience and Natural Resources, University of Copenhagen, Frederiksberg C, 1958, Denmark

*Correspondence to*: Christian Juncher Jørgensen (cjj@bios.au.dk)

**Abstract.** In this paper, the performance of a low-cost and low-power methane ($CH_4$) sensing system prototype based on a metal oxide sensor (MOS) sensitive to $CH_4$ is tested in a natural $CH_4$ emitting environment at the Greenland Ice sheet (GrIS). We investigate if the MOS could be used as a supplementary measurement technique for monitoring $CH_4$ emissions from the GrIS with the scope of setting up a $CH_4$ monitoring network along the GrIS. The performance of the MOS is evaluated on basis of parallel measurements using a CRDS reference instrument for $CH_4$ over a field calibration period of approximately 100 h. Results from the field calibration period show that $CH_4$ concentrations measured with the MOS is in very good agreement with the reference CRDS. The absolute concentration difference between the MOS and the CRDS reference values within the measured concentration range of approximately 2-100 ppm $CH_4$ were generally lower than 5 ppm $CH_4$, while the relative concentration deviations between the MOS and the CRDS were generally below 10 %. Calculated mean bias error for the entire field calibration period was -0.05 ppm with a standard deviation of ± 1.69 ppm (n = 37,140). The results confirms that low-cost and low-power MOS can be effectively used for atmospheric $CH_4$ measurements under stable water vapor conditions. The primary scientific importance of the study is that it provides a clear example on how the application of low cost technology can enhance our future understanding on the climatic feedbacks from the cryosphere to the atmosphere.

## 1. Introduction

Constraining the various sources and sinks in the global methane ($CH_4$) budget is becoming an increasingly important parameter in mitigating climate change (Saunois et al., 2016). While the Arctic is generally considered a major global emitter of $CH_4$ to the atmosphere, significant uncertainty exists to the seasonal dynamics and strength of both $CH_4$ sources and $CH_4$ sinks from both terrestrial and marine environments, as well as the cryosphere (Callaghan et al., 2011; Emmerton et al., 2014; Juncher Jørgensen et al., 2015; Pirk et al., 2017; Zona et al., 2016). Recently, a previously unknown source of $CH_4$ emission to the atmosphere was identified where $CH_4$ is emitted from meltwater originating in the subglacial domain of the Greenland Ice Sheet (GrIS) (Christiansen and Jørgensen, 2018; Lamarche-Gagnon et al., 2018; Wadham et al., 2019). The spatiotemporal



coverage of the new $CH_4$ source is yet to be determined and the overall climatic importance of this new component in the Arctic $CH_4$ budget is still unknown. Future studies are needed in order to assess the overall climatic significance of this source of $CH_4$ emission from the cryosphere to the atmosphere. The current state of knowledge on the $CH_4$ exchange from Greenland

is inherently limited by the remoteness of many field sites with following high expedition cost and limitations to the spatial coverage and temporal duration of field measurements. Adding to this is the financial and logistical challenges of bringing high precision analyzers into the field, keeping them powered, running and shielded in the harsh environments often encountered in the Arctic. Thus, there is substantial potential and need to develop low-power techniques and measurement systems that can perform reliable autonomous $CH_4$ concentration measurements. The emergence of low-cost/low-power sensor

technology in recent years provides an opportunity to overcome many of current restraints on obtaining continuous field measurements from a wide range of natural $CH_4$ emitting systems (wetlands, ice sheets, marine gas seeps, lakes, permafrost) and expand the network of continuous measurements in remote areas maximizing our understanding of these systems and minimizing the risk of losing valuable analytical equipment.

Low-cost metal oxide gas sensors (MOS) have been widely used for sensing various gases under atmospheric conditions (Wang et al., 2010). However, MOS sensors have significant obstacles to their direct use as air quality monitors as their output signal is influenced by the concentrations of both the target and interfering gases, as well as the temperature and humidity effects (Masson et al., 2015; Sohn et al., 2008). Other known challenges to the use of MOS are baseline drift over time, caused by either changes in the heat output of the sensing element or due to poisoning of the sensor surface (Peterson et al., 2017).


In recent years, investigations into the performance of $CH_4$ sensitive MOS sensors for the measurement of atmospheric $CH_4$ have been made under both natural and controlled conditions (van den Bossche et al., 2017; Eugster and Kling, 2012; Penza et al., 2015). These studies have been prompted by an increased interest in finding effective methods to quantify $CH_4$ emissions to the atmosphere from both natural systems and man-made systems such as landfills or biogas production plants. Using sensor

specific post-processing to compensate for variations in relative humidity and air temperature, the previous studies have demonstrated a high potential for the low-cost and low-power monitoring of $CH_4$ concentrations above the atmospheric background level for various applications and in sensor network grids. In the current study, we have in situ tested the performance of a $CH_4$ sensitive MOS (Figaro TGS2611-E00) against a state-of-the-art cavity ring-down spectrometer for $CH_4$ (Ultra-portable Greenhouse Gas Analyzer, Los Gatos Research Inc.) to measure $CH_4$ concentrations in the air expelled from a

subglacial meltwater outlet at GrIS. This was done to assess the MOS's potential for serving as a sensing element in future studies of the important scientific knowledge gap concerning the climatic feedbacks following $CH_4$ emissions from the subglacial domain under the Greenland Ice Sheet to the atmosphere.



## 2. Materials and methods


### 2.1 Field site and instrumentation

The study site is located on the southern flank at the terminus of the Isunnguata Sermia Glacier at the western margin of the GrIS (67°09'16.40''N 50°04'08.48''W) at an elevation of 450 meter above sea level (Fig. 1). At a small subglacial meltwater discharge outlet in this area, we performed measurements of $CH_4$ concentrations in the subglacial air expelled from naturally
occurring caves carved out by meltwater below the ice sheet. The measurements were done in the period between June 22nd and 26th 2018. A more detailed description of the study site at the GrIS is given in (Christiansen and Jørgensen, 2018).

To sample the subglacial air the sampling tube was attached to an aluminum pole inserted approximately 5 meters into an ice cave with the inlet of the sampling tube connected to a 100 ml water trap (Fig. 2a). At the end of aluminum pole inside the
subglacial cave the humidity and temperature of the subglacial air was measured every 10 seconds with a combined sensor (S-THB-M008, Onset, USA) connected to a datalogger (U30, Onset, USA). A sampling tube of 50 meter (inner diameter of 4 mm and total volume of 630 mL) was connected to the inlet of the CRDS (Ultraportable Greenhouse Gas Analyzer, Los Gatos Research, USA) for real-time reference concentration analysis of $CH_4$, carbon dioxide ($CO_2$) and water vapor ($H_2O$) (Fig. 2a). The diaphragm pump of the CRDS creates a constant flow of 800 mL min$^{-1}$. Through a 1 meter tube the outlet flow (800 mL
min$^{-1}$) of the CRDS constantly flushed an open-ended enclosure (400 mL) in which the prototype $CH_4$ sensing system (MOS) was inserted (Fig. 2b). During the field calibration period (22nd to 26th July 2018), the rapid flushing of the air volume in the enclosure with the MOS system (2 times per minute) and the non-destructive sampling principle of the CRDS collectively ensured that the concentration of $CH_4$ in the above the MOS sensor was identical to the CRDS at the same time step.

Following the field calibration test of approximately 100 h, the MOS system was left as an autonomous monitoring system where a constant air-flow of approximately 3 L min$^{-1}$ to the enclosure with the MOS system was supplied by a 12 volt diaphragm pump (Thomas pumps, 1410VD DC) instead of using the outlet from the CRDS. During this period the system was powered by 12V LiFePO$_4$ batteries connected to solar panels and a voltage regulator. During the autonomous measuring period the MOS system and enclosure was placed in a water-proof case and buried under a pile of rocks to minimize the impact of
sunlight induced temperature variations of the sensor system allowing a direct comparison between $CH_4$ concentration measurements of the CRDS and MOS systems.





## 2.2 The MOS system

The MOS system (Fig. 2c) consists of a microcontroller (Arduino Uno) and datalogger shield (DeekRobot data logging shield

V1.0) holding the board-mounted metal oxide $CH_4$ sensor (Figaro TGS 2611-E00) and an additional temperature/relative humidity micro sensor (GY-21 HTU21). Logging frequency of the CRDS and MOS was 1 and 10 seconds, respectively. The $CH_4$ sensitive MOS consists of a tin(IV)oxide ($SnO_2$) semiconductor which has low conductivity in clean air. In the presence of $CH_4$, the sensor's conductivity increases depending on the gas concentration in the air (Kumar et al., 2009). A simple electrical circuit convert the changes in conductivity at the sensing element as the gas concentrations vary to a change in output

voltage across the voltage divider (see Fig. 3). Both the heater and the sensing circuit of the MOS was powered by the 5 volt regulated output of the Arduino Uno. The analogue output of the MOS was connected to the 10-bit analogue input on the Arduino Uno using a 10 kOhm precision load-resistor in the voltage divider.

## 2.3 Laboratory calibration of the MOS sensor

In preparing for the field test of the $CH_4$ sensing system prototype, the MOS was performance tested and calibrated in a controlled laboratory environment to evaluate both the response time to variations in methane concentration in the concentration range 0-100 ppm $CH_4$ at three different levels of relative humidity ($37\pm2$ %, $55\pm3$ % and $76\pm3$ %). Synthetic air (80 % $N_2$ and 20 % $O_2$) was used as zero gas for the laboratory test to which various concentrations of a $CH_4$ containing span gas was mixed in using a HovaCAL calibration gas generator (IAS Gmbh, Germany). After mixing of the zero gas and span

gas, the calibration gas was humidified using a water filled impinger similar to (van den Bossche et al., 2017). At each humidity level, the output voltage from the MOS was logged using a Campbell CR1000 datalogger at a 2 second sampling frequency. A pre-programmed calibration step sequence was used for all three humidity levels, consisting of time steps of each 10 minutes in which the sensor was exposed to either zero gas or a calibration gas mixture in the applied the concentration range in an alternating step pattern (Fig. 4). The temperature in the laboratory, zero gas, mixed calibration gas and water in impinger was

kept constant around 22 °C throughout the calibration test.

The sensor resistance ($R_O$) at exposure to the $CH_4$ free reference gas can be calculated at each of the three different humidity levels according to Eq. (1):

$$R_0 = \frac{V_C * R_L}{V_{OUT}} - R_L \, ,\qquad\qquad(1)$$

where $V_C$ is the circuit voltage (i.e. 5 volt DC), $R_L$ is the load resistance (10 kOhm) and $V_{OUT}$ is the measured output voltage (see also Eugster and Kling (2012) for further description).





The sensor resistance at various calibration gas concentrations ($R_S$) at different concentration steps in the calibration sequence
can also be calculated using equation 1 for each of the three humidity levels (i.e. Rs replaces Ro in equation 1). For the tested
type of MOS, the sensor resistance ratio ($R_S/R_O$) between the sensor resistance at a given concentration level (Rs) and the
sensor resistance at the reference level (Ro) is inversely proportional to the absolute $CH_4$ concentration and can be modelled
using e.g. a power fit (Fig. 5).

### 2.4 Field-calibration of the MOS

Field calibration of the MOS was done at the meltwater outlet at the Greenland Ice Sheet by parallel measurements of the same
air mass by the MOS sensor system and a state-of-the-art CRDS. Since access to a controlled, humidified zero gas was not
possible in the field, the output of the MOS when exposed to ambient air at the atmospheric background concentration of $CH_4$
(approximately 1.9 ppm) close to the ice sheet was used to calculate the average ambient sensor resistance ($R_{0*}$) using Eq. 1,
which was then used to establish the resistance ratio ($R_S/R_{0*}$) vs. $CH_4$ concentration field calibration function for the MOS
(Fig. 6).

### 2.5 Data processing

The measured raw time series data from the MOS were smoothed using simple exponential smoothing according to Eq. (2):

$$s_t = \alpha x_t + (1-\alpha)s_{t-1} \quad \text{for } t>0 \tag{2}$$

where $s_t$ is the smoothed $CH_4$ concentration value (ppm), $\alpha$ is the smoothing factor and $s_{t-1}$ is the previous smoothed $CH_4$
concentration value (ppm). At time zero (t=0), the $s_t$ is equal to the first unsmoothed raw $CH_4$ value of the MOS. The optimum
value for $\alpha$ was determined using Microsoft Excel solver, by minimizing the total average root mean square error (RMSE)
between the raw data from the MOS and the simultaneous concentration measurement of the CRDS. Results show an optimal
value of 0.042, which for the sake of simplicity was rounded to 0.05, and subsequently used for both the CRDS and MOS data
series (Fig. 7).



# 3. Results and Discussion

## 3.1 Laboratory calibration test of the MOS

Fig. 3 shows the relationship between the resistance ratio ($R_S/R_O$) for the step test at three humidity levels, where Ro is calculated for each humidity levels based on the average voltage output of the sensor in the time steps where it was exposed to the $CH_4$-free synthetic air. It is observed that a near identical response function can be obtained across the three different water vapor concentrations in the air, as long as the water concentration of the zero gas is the same as in the span gas. Based on existing knowledge of the expected air temperature variations at the in situ sampling point at the GrIS (Christiansen and Jørgensen, 2018), the humidity calibration was only carried out at a single temperature in this study. However, variations in the ambient air temperature is also expected to have a linear scaling effect for the type MOS system tested in this study (van den Bossche et al., 2017).

## 3.2 Field-calibration of the metal oxide sensor

The measured $R_S/R_{O*}$ ratios per time step over the field calibration period were converted into absolute $CH_4$ concentrations using the regression statistics of the applied power model (Fig. 6). While the same regression model equations are applied in both the laboratory calibration and field calibration, significant deviation in the model parameters are observed between the laboratory calibration as a group and the field calibration. The reason for this difference is unknown, but a possible explanation could be the potential difference in input heater voltage for the MOS sensor (i.e. pin 1 and 4 in Fig. 1) which has been reported to linearly scale the $CH_4$ concentration measurements study (van den Bossche et al., 2017). In the laboratory test, the heater circuit of the MOS was supplied by the 5 volt regulated output from the CR1000 datalogger, whereas the heater circuit was supplied from the Arduino Uno's 5 volt regulated output. Future test should aim to investigate if the differences between the results from the laboratory and field calibration can be minimized by using the same type of datalogger and identical power supply (fx. Rechargeable lithium ion battery pack) both in the laboratory and in the field. Results from this type of test could reveal if field calibration for each individual MOS system is needed, or if batch calibrations of several identical MOS-system can be performed in the laboratory without the need for time-consuming field calibration.

## 3.3 Time-series plot of $CH_4$ concentration from reference CRDS and MOS

Due to the dynamic environment at the margin if the GrIS, the physical configuration of the sampling point will vary both over the melt-season as well as on an inter-annual basis. In our previous study, high frequency variations in $CH_4$ concentrations in the subglacial air were observed in a downward draping curve style where a high concentration plateau was interrupted by





rapid decreases in $CH_4$ concentration (Christiansen and Jørgensen, 2018). This pattern was interpreted to be an effect of micro-turbulent and wind driven dilution of the sample gas in the ice cave by atmospheric air with a $CH_4$ concentration of approximately 1.9 ppm. In the current study, exponential smoothing of the raw values is used to compensate for the potential effects of physical disturbance of the sample gas caused by wind driven turbulent mixing of atmospheric background air at the subglacial sample point. Also, temporal smoothing can compensate for some of the sensor specific variation in response time

improving the pairwise measurement comparability between the CRDS and the MOS. According to the manufacturer, the CRDS is specified to have a response time of less than 1 hz, while the response time of the MOS is expected to be slower. The $T_{90}$ response time for a similar $SnO_2$-based $CH_4$ sensor separated with a thin silicone membrane has been reported to be between 1-30 minutes (Boulart et al., 2010), for which the shorter time range is comparable to what is observed in the laboratory calibration of this study (Fig. 4).


   The time series plot of the raw and exponentially smoothed $CH_4$ data from the CRDS ($CRDS_{smooth}$) is shown together with the pairwise error between the raw data and the smoothed data (Fig. 7a). Similarly, the time series plot of the raw and exponentially smoothed $CH_4$ data from the MOS ($MOS_{smooth}$) is shown together with the pairwise error between the raw data and the smoothed data (Fig. 7b). It is generally observed that over the first 4 days of the calibration test, very low differences are

observed between the raw data $CH_4$ concentration and the smoothed $CH_4$ concentrations for both $CRDS_{smooth}$ and $MOS_{smooth}$, with absolute errors below 5 ppm (Fig. 7a & 7b). At the end of the field calibrations, higher errors are observed following the larger spread in $CH_4$ concentration measurements of both the CRDS and the MOS. CRDS analyzers across different brands and manufacturers generally perform very consistently and have a highly linear measurement response across the effective concentration range without any tendencies for increasing analytical error with increasing gas concentrations (Brannon et al.,

2016). Fluctuations in $CH_4$ concentrations in the subglacial air were also observed in (Christiansen and Jørgensen, 2018) using a CRDS from another manufacturer (G4301 GasScouter, Picarro Inc.). These variations were attributed to the dynamic and micro-turbulent environment in the subglacial cavity were the gas concentrations were measured and are likely produced by both air movement generated by the shear stress of the running meltwater as well as turbulent intrusion of atmospheric air generated by shifting winds speeds at the measurement location at the ice margin.


   According to the field notes for the current study, a shift in overall wind regime took place during the 25th of June 2018, where the weather shifted from calm and sunny conditions to more windy conditions dominated by strong catabatic easterly winds coming off the GrIS. A best estimate of the overall time period where more windy conditions occurred during the field calibration period is indicated with grey background in Fig. 7. Unfortunately, no direct measurements of wind movement were

made during the fieldwork period at the location. Measurements of air temperature at the sample inlet point in the subglacial cavity (Fig. 7c) shows that an initial period with diurnal temperature variations of approximately 0.1 to 0.2 °C was followed by a period with more fluctuating temperature variations of up to + 0.6 °C. The period with higher variability corresponds to the period where higher winds speeds predominate and the deviations between the raw and smoothed $CH_4$ are the greatest.





The higher variability in air temperature measurements during the more windy weather is interpreted as being a product of

more turbulent wind conditions right at the margin of the GrIS and opening to the subglacial cavity by which higher amounts of warmer atmospheric air with an approximate $CH_4$ concentration of approximately 1.9 ppm is introduced into the subglacial cavity. The introduction of these atmospheric air masses results in both short-term temperature increases as well as dilution of the subglacial $CH_4$ concentration in the cavity producing the more variable $CH_4$ concentration patterns observed in both the CRDS and MOS raw data. In the absence of direct measurements of wind speed and micro-turbulence at the margin of the ice,

rapid variations in air temperature at the sample inlet point with an amplitude greater than the 0.2 ºC appear as a reasonable indicator or proxy for micro-turbulent dilution and physical disturbance of the source signal, which can effectively be filtered out by the application of exponential smoothing.

The relative error between each $MOS_{smooth}$ and $CRDS_{smooth}$ measurement pair can be expressed as the percentage that the

difference constitutes compared to the reference CRDS concentration (i.e $MOS_{smooth}$ – $CRDS_{smooth}$/$CRDS_{smooth}$ x 100). It is seen that the pairwise relative error between the $MOS_{smooth}$ and $CRDS_{smooth}$ shows similar non-systematic variations in both the calm weather and windy time period with relative errors typically below ± 10% (Fig. 7d). This result show both that the accuracy of the $CH_4$ concentration measured by the MOS are in close agreement with the reference CRDS and that the exponential smoothing effectively compensates for short term physical disturbances at the measurement point. The result also

indicate that no systematic drift or over/underestimation is apparent when comparing the $MOS_{smooth}$ to the $CRDS_{smooth}$ over the 100 h field calibration period (Fig. 7b). When considering the magnitude of the absolute errors between the raw and smoothed $CH_4$ concentration for both the CRDS and the MOS, together with the temporal pattern in the development of the relative error, it shows that the high frequency concentration fluctuations measured with the MOS are most likely the product of physical disturbances at the measurement point (primary sampling error), and not by an analytical error introduced by the

MOS itself (secondary sampling error).

As supplement to the pairwise error comparison, average time-series performance statistics for the difference between the $MOS_{smooth}$ and $CRDS_{smooth}$ time series can be calculated for both the full field calibration period, as well for the non-turbulent time period with limited observed physical disturbance at the sampling point (Table 1). Mean bias errors for both the non-

turbulent and full time series are approximately ± 0.01 ppm $CH_4$ with standards deviations of ± 1.3 to 1.7 ppm $CH_4$ respectively.

### 3.4 Post-correction and cross-interference evaluation

One of the main obstacles previously reported concerning the use of MOS's for monitoring of gases in ambient air is the possible effect of variations in air temperature and humidity in the sampling environment (Masson et al., 2015; Sohn et al.,

2008). One approach to compensate for this potential measurement error is to post-correct for variations in temperature and humidity, based on either generic temperature and humidity dependency curves supplied by the sensor company (Eugster and





Kling, 2012) or by performing sensor calibrations under controlled levels of temperature and humidity in the laboratory (van den Bossche et al., 2017).

Measurements from the air-filled cavity under the GrIS document a very stable sampling environment with a relative humidity throughout the sampling period of close to 100 % RH (data not shown) and only minor air temperature variations between approximately 0.05 °C during the night and 0.25 °C during mid-day (Fig. 7d). Because of these stable and well-buffered environmental conditions, no post-corrections due to variations in temperature and relative humidity are evaluated to be necessary for this particular sampling environment.


Observed variations in maximum air temperature in the subglacial cavity correspond to field observation of the time of the day when maximum meltwater discharge occurs. We assume that the observed temperature pattern reflects the impact of thermal heat diffusion from this running meltwater to the air immediately above, but would need direct measurements of the daily variations in meltwater temperature to verify this assumption.


The emitted $CH_4$ may originate from both thermogenic and/or biogenic sources below the GrIS. If the primary source of $CH_4$ is thermogenic, the emission may also be accompanied by more complex hydrocarbons, including ethane ($C_2H_6$), while this will not be the case if the source is biogenic (Hopkins et al., 2016). Since the MOS used in the study is non-selective to $CH_4$ due to its basic principle of operation (Eugster and Kling, 2012; Wang et al., 2010), the presence of other hydrocarbons such

as ethanol ($C_2H_6O$), isobutene ($C_4H_{10}$) and potentially also other low molecular weight alkanes could potentially cause cross-interference with the $CH_4$ measurement. It follows, that if the source of the $CH_4$ that is emitted for the subglacial domain originates in thermogenic natural gas reservoirs under the GrIS, the other non $CH_4$-hydrocarbons could potentially affect the measurements performed by the MOS, while passing non-detected by the CRDS. However, since the magnitudes and temporal patterns in $CH_4$ concentrations are similar in both the CRDS and MOS it is assumed that the gases emitted from the subglacial

domain are primarily $CH_4$ and $CO_2$ with very limited potential for cross-interference from other hydrocarbon gases. Also, isotopic analysis of the emitted $CH_4$ and $CO_2$ in (Lamarche-Gagnon et al., 2018) as well as unpublished data from this study, have shown that the emitted $CH_4$ is dominantly of microbial origin and has isotopic similarity to $CH_4$ produced by anaerobic decomposition of organic carbon in wetlands. It is therefore assumed that there is no need for any post correction of the $CH_4$ concentrations measured by the MOS in this type of environment due to lack of cross-interference from other hydrocarbon

gases.

## 3.5 Autonomous $CH_4$ monitoring using MOS system

The combined time period in which $CH_4$ concentrations were measured can be divided into three separate periods depending on the analytical devices used, namely period 1 corresponding to the field calibration period where both the CRDS and MOS



were in operation (approximately 100 h), period 2 where only the CRDS was in operation (approximately 24 h) and period 3 where only the MOS was in operation (Fig. 8). Continuous $CH_4$ data from period 3 exist for the period 27th June to 15th July 2018. When comparing the combined $CH_4$ concentration curves from all three periods it is observed that the $CRDS_{smooth}$ and $MOS_{smooth}$ follow each other as described above (Fig. 7). $CRDS_{smooth}$ data for period 2 fills the data gap between the MOS measurement of period 1 and 3, where the $MOS_{smooth}$ concentration data departs very close to the concentration level where

the $CRDS_{smooth}$ measurements end. Due to the nature of the study design and difficult access to the remote field site at the GrIS, the accuracy and precision of the $MOS_{smooth}$ cannot be evaluated for the period 3 where only the MOS system was operating. However, the pattern in which subglacial $CH_4$ concentrations varied and the estimated minimum and maximum values appear similar to the values of the calibration period. When comparing the complete time period of this study to e.g. Eugster and Kling (2012), no significant sensor drift is expected over the monitoring time period. Additional and extended field work at the GrIS

with repeated calibration at the end of the field deployment period is needed to quantify the potential sensor drift, as well as stability range over longer time scales. Nonetheless, the observed performance of the MOS during the calibration period with ppm-level accuracies and subsequent trouble-free operation running as an autonomous unit shows that this type of low-cost and low-power $CH_4$ sensing system holds a great potential for the further development and refinement of a greater sensor network at representative meltwater outlets at the Greenland ice Sheet. The major scientific scope of this performance test is

that we can demonstrate a realistic low-cost technical solution for closing one of the most critical knowledge gaps in the Arctic carbon budget, namely the climatic impact of $CH_4$ emissions to the atmosphere under both current and future warmer climatic conditions.

The next steps and lessons learned from this study deals with the further development of the low-power system for actual $CH_4$

emission measurements, which involves measurements of air volume and meltwater discharge as well as continuous measurements of the dissolved $CH_4$ in the meltwater, similar to (Lamarche-Gagnon et al., 2018). Also, optimizing the positioning of gas sensing equipment at the measurement point should be done to reduce the potential physical disturbances due to micro-turbulence and intrusion of atmospheric air in the subglacial cavity. Furthermore, an improved adjustment scheme should be developed to account for the dynamic melt back of the ice margin over the melt season, which requires either manual

or automated sample point relocation to keep the sampling point at an optimal physical location. Finally, more work is needed to test what modification to the systems are needed to establish a universal calibration curve in the laboratory, so that the need for field calibration with the reference CRDS can be eliminated.



## 4. Conclusions

Recent discoveries at the Greenland Ice Sheet (GrIS) have revealed a so far overlooked source of $CH_4$ from the subglacial domain under the ice to the atmosphere. Development of low-power $CH_4$ monitoring systems based on low cost metal oxide sensors (MOS) could enable the development of a sensor network at representative meltwater outlets at the GrIS which significantly could enhance the fundamental understanding of the phenomena's climatic importance. In the current study, the performance of a metal oxide sensor sensitive to $CH_4$ was tested in an air-filled cavity at the edge of the Greenland Ice Sheet

over an initial field calibration period of approximately 100 h using both a reference gas analyzer based on cavity ring-down spectroscopy (CRDS) and a low-cost metal oxide sensor (MOS) followed by a period of autonomous $CH_4$ concentration monitoring using only the MOS system. Parallel measurements using a common inlet show good agreement between the MOS and the CRDS over time under the stable environmental conditions under the ice. Exponential smoothing of the raw data from both the CRDS and MOS effectively remove high frequency concentration variations induced by physical disturbance of the

air in the subglacial cavity under more turbulent wind conditions at the margin of the ice sheet. Based on concentration values of the smoothed CRDS and MOS data, the pairwise measurement errors were generally below ± 5 ppm $CH_4$ between the MOS and the CRDS reference value. Pairwise relative errors were generally below ± 10 % between the MOS and the CRDS reference value. The mean bias error for the entire field calibration period was -0.05 ppm $CH_4$ with a standard deviation of ± 1.69 ppm $CH_4$. If only data for the non-turbulent time period was evaluated, the mean bias error was 0.09 ppm $CH_4$ with a standard

deviation of ± 1.35 ppm $CH_4$. Due to the very clean and stable sampling environment in the air-filled cavity under the Greenland Ice Sheet, no post-corrections for variations in air temperature, humidity or cross interference from other hydrocarbon gases were needed for the MOS measurements. Combined with measurement of airflow and meltwater discharge, the measurement of $CH_4$ concentrations can be used for determination of the mass flux of $CH_4$ to the atmosphere. The study demonstrates a clear potential for expanded monitoring of spatial and temporal variation in $CH_4$ emissions from the subglacial

domain of the Greenland Ice Sheet using low-cost and low-power MOS.

## 5. Author contribution

CJJ and JRC designed and carried out the field experiments. CJJ, JMO and KFU planned and carried out the laboratory calibrations. CJJ and JRC prepared the manuscript with contributions from all co-authors.

## 6. Acknowledgements

This work was supported by a research grant from the "Brødrene Hartsmanns Fond" for the project "Udledning af metan til atmosfæren fra gletchere" and performed as part of the "Arctic Research Centre" and "iClimate" research frameworks at Aarhus University. Laboratory test of MOS sensors was performed by FORCE Technology through support from the Danish Agency for Innovation as part of the project "Den Danske Renluftsektor".



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





**Figure captions**


Figure 1: Overview of the sampling location at Isunnguata Sermia Glacier at the western margin of the Greenland Ice sheet during June 2018. (a) Location of sampling region the island of Greenland, (b) regional location of the outlet glacier, (c) location of the meltwater outlet at Isunnguata Sermia and (d) local sample location with investigated subglacial cavity marked with red circle. Source of (a), (b) and (c): Google Earth, earth.google.com/web/.


Figure 2: (a) Close-up of air-filled cavity below the Greenland Ice Sheet next to the lateral meltwater outlet. The aluminum pole extends approximately 5 meters into the cavity and holds the inlet tube and the temperature and humidity smart sensor. (b) Conceptual diagram of the MOS system placement in an enclosure constantly flushed by the outflow from the CRDS analyzer (c) Close-up of the board mounted MOS and temperature/humidity micro sensor. The MOS system consisted of 1)

a microcontroller, 2) Datalogger shield holding metal oxide $CH_4$ sensor and 3) an additional temperature/relative humidity micro sensor.

Figure 3: Simplified schematic of the metal oxide sensor (MOS) system consisting of a TGS2611-E00 with pin 3 and 4 connected to the 5-volt output of the Arduino Uno, pin 1 connected to ground and pin 2 connected to the analogue input of

the Arduino Uno and a 10kOhm load resister, which also connects to ground.

Figure 4: Outlet voltages of the MOS during laboratory step calibration at stabilized levels of relative humidity ($37 \pm 2$ %, $55 \pm 3$ % and $76 \pm 3$ %) in both the zero and span gas at alternating concentration of $CH_4$ in the calibration gas between 10 and 100 ppm $CH_4$. Each time step lasted 10 min and sequences with grey shadings show time periods where the sensor was

exposed to $CH_4$ free zero gas.

Figure 5: Resistance ratio of MOS as three levels of relative humidity at $CH_4$ concentration levels between 10 to 100 ppm $CH_4$ in humidified synthetic air.

Figure 6: Regression plot of calculated MOS resistance ratio $R_S/R_{O^*}$ vs. the reference in situ $CH_4$ concentrations from the CRDS.

Figure 7: (a) Grey dots show raw $CH_4$ concentration from CRDS. Black line show CRDS $CH_4$ concentration values following exponential smoothing. Black bars show absolute error between raw and smoothed values. (b) Grey dots show

calculated raw $CH_4$ concentration from MOS. Black line show MOS $CH_4$ concentration values following exponential smoothing. Black bars show absolute error between raw and smoothed values. (c) Black dots show temperature of air in subglacial cavity. (d) Black bars show the relative error in percentage between the $MOS_{smooth}$ and $CRDS_{smooth}$ divided by the $CRDS_{smooth}$ concentration. Grey background shading indicates period with higher observed turbulence at the margin of the GrIS.


Figure 8: Smoothed time series measurements of $CH_4$ at the Greenland Ice Sheet using both the cavity ring-down spectroscopy (CRDS) reference monitor and the metal oxide sensor (MOS).

Table 1. Statistics for the calculated differences between the smoothed MOS and smoothed CDRS data series in both the

non-turbulent time period and full field calibration period. The unit for error and difference values is ppm.



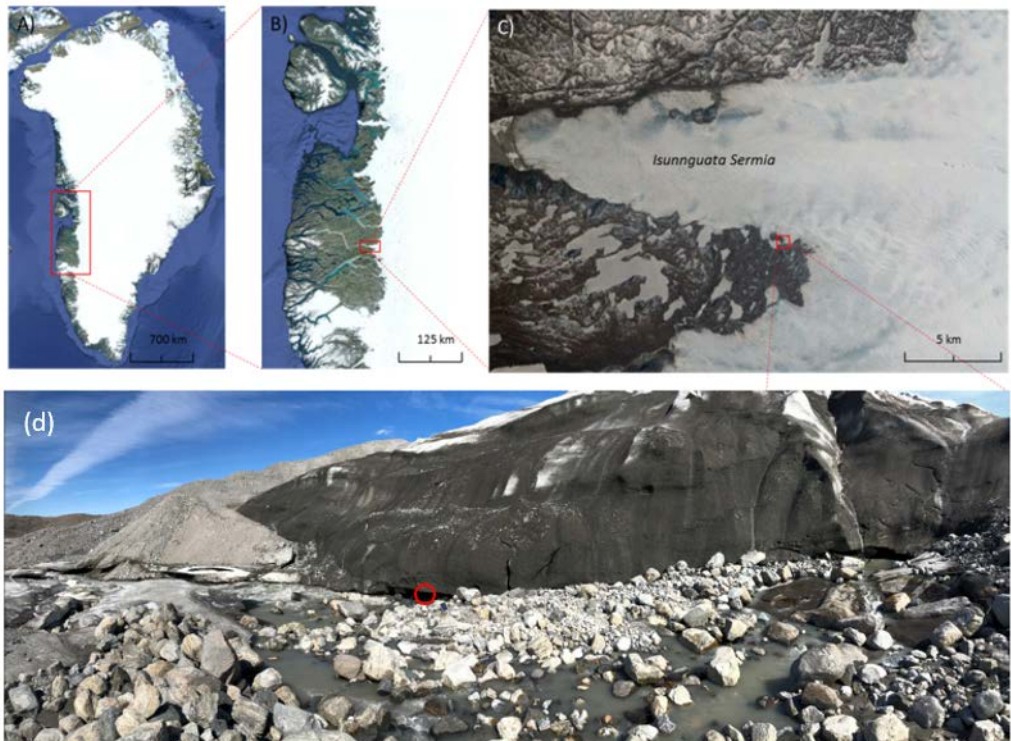

Figure 1: Overview of the sampling location at Isunnguata Sermia Glacier at the western margin of the Greenland Ice sheet during June 2018. (a) Location of sampling region the island of Greenland, (b) regional location of the outlet glacier, (c) location of the meltwater outlet at Isunnguata Sermia and (d) local sample location with investigated subglacial cavity marked with red circle. Source of (a), (b) and (c): © Google Earth, earth.google.com/web/.



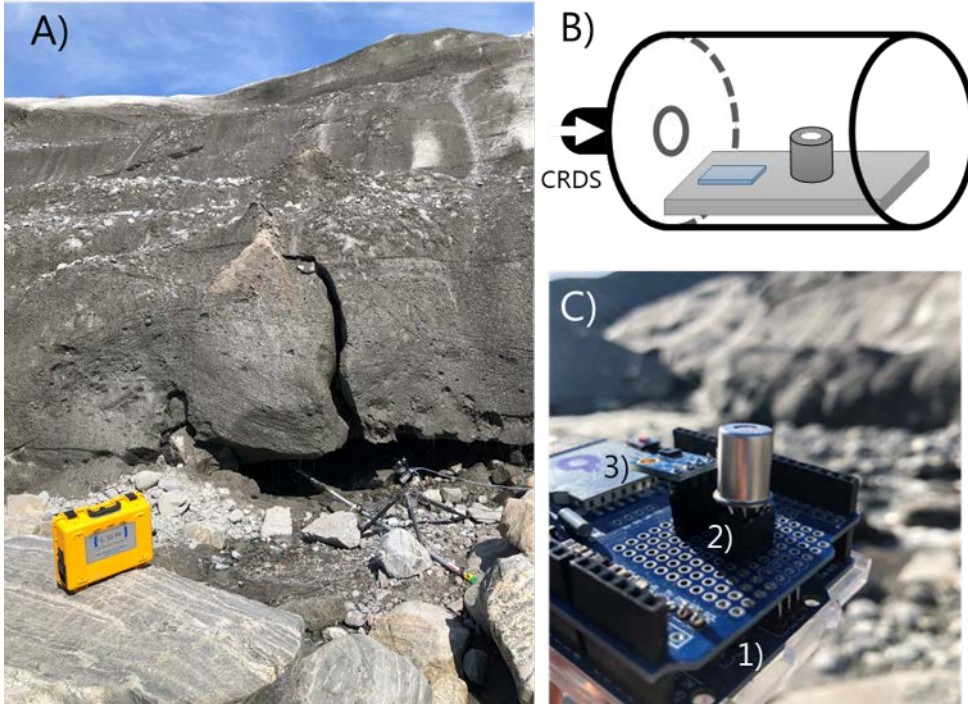

Figure 2: (a) Close-up of air-filled cavity below the Greenland Ice Sheet next to the lateral meltwater outlet. The aluminum pole extends approximately 5 meters into the cavity and holds the inlet tube and the temperature and humidity smart sensor. (b) Conceptual diagram of the MOS system placement in an enclosure constantly flushed by the outflow from the CRDS analyzer (c) Close-up of the board mounted MOS and temperature/humidity micro sensor. The MOS system consisted of 1) a microcontroller, 2) Datalogger shield holding metal oxide $CH_4$ sensor and 3) an additional temperature/relative humidity
micro sensor.





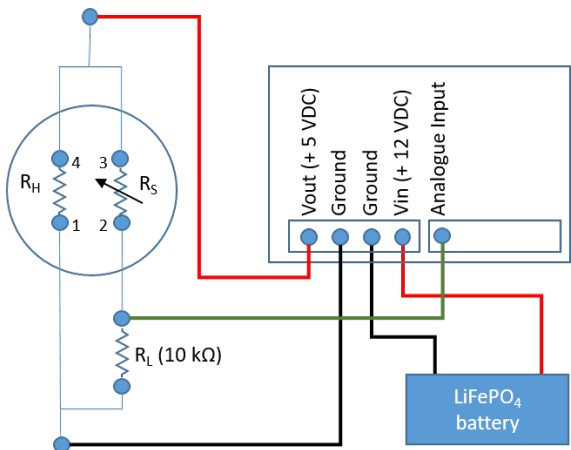

Figure 3: Simplified schematic of the metal oxide sensor (MOS) system consisting of a TGS2611-E00 with pin 3 and 4 connected to the 5-volt output of the Arduino Uno, pin 1 connected to ground and pin 2 connected to the analogue input of the Arduino Uno and a 10kOhm load resister, which also connects to ground.





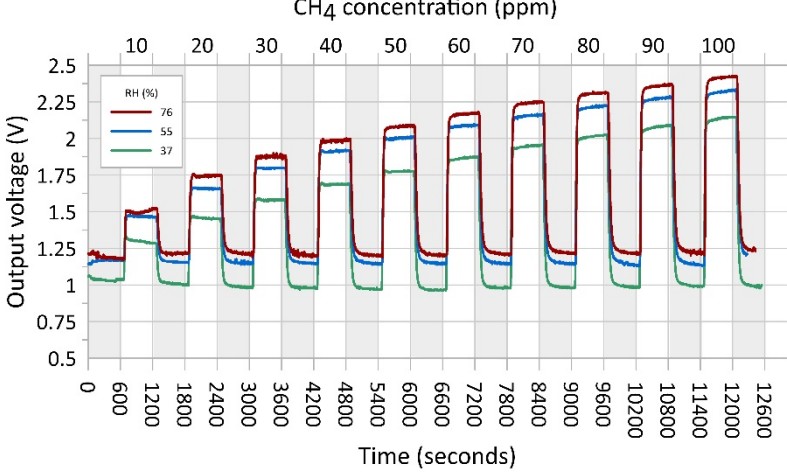


Figure 4: Outlet voltages of the MOS during laboratory step calibration at stabilized levels of relative humidity ($37 \pm 2$ %, 55 $\pm 3$ % and $76 \pm 3$ %) in both the zero and span gas at alternating concentration of $CH_4$ in the calibration gas between 10 and 100 ppm $CH_4$. Each time step lasted 10 min and sequences with grey shadings show time periods where the sensor was exposed to $CH_4$ free zero gas.






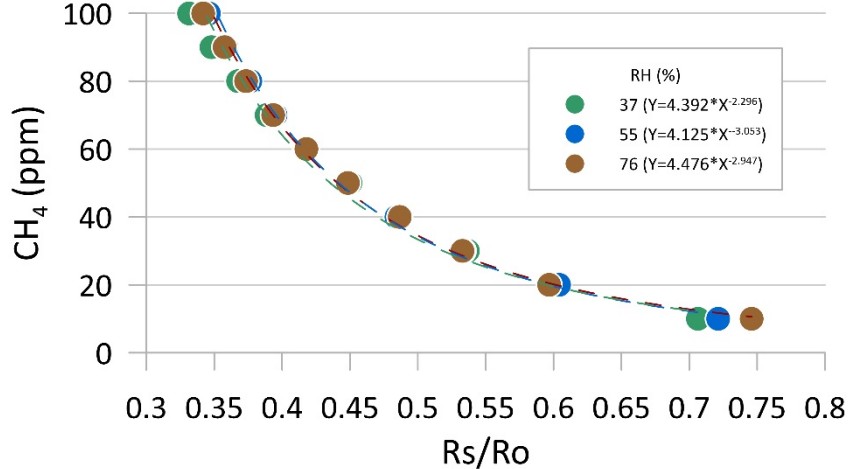

Figure 5: Resistance ratio of MOS as three levels of relative humidity at $CH_4$ concentration levels between 10 to 100 ppm
$CH_4$ in humidified synthetic air.





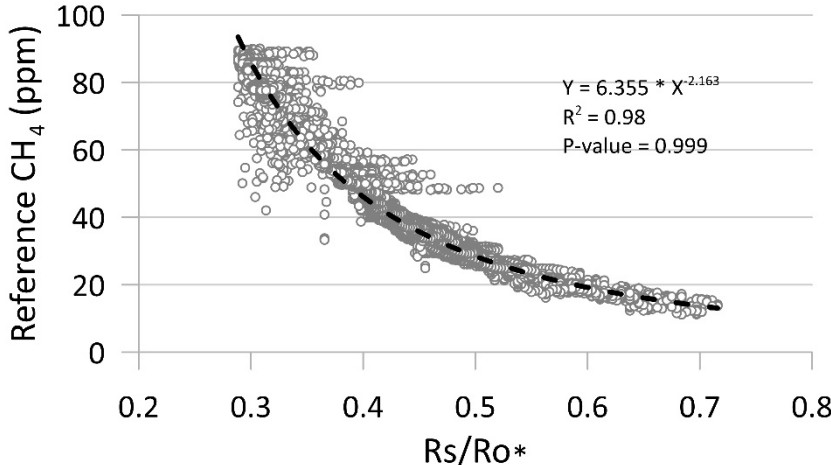

Figure 6: Regression plot of calculated MOS resistance ratio $R_S/R_{O*}$ vs. the reference in situ $CH_4$ concentrations from the CRDS.



Figure 7: (a) Grey dots show raw $CH_4$ concentration from CRDS. Black line show CRDS $CH_4$ concentration values following exponential smoothing. Black bars show absolute error between raw and smoothed values. (b) Grey dots show calculated raw $CH_4$ concentration from MOS. Black line show MOS $CH_4$ concentration values following exponential smoothing. Black bars show absolute error between raw and smoothed values. (c) Black dots show temperature of air in subglacial cavity. (d) Black bars show the relative error in percentage between the $MOS_{smooth}$ and $CRDS_{smooth}$ divided by the $CRDS_{smooth}$ concentration. Grey background shading indicates period with higher observed turbulence at the margin of the GrIS.





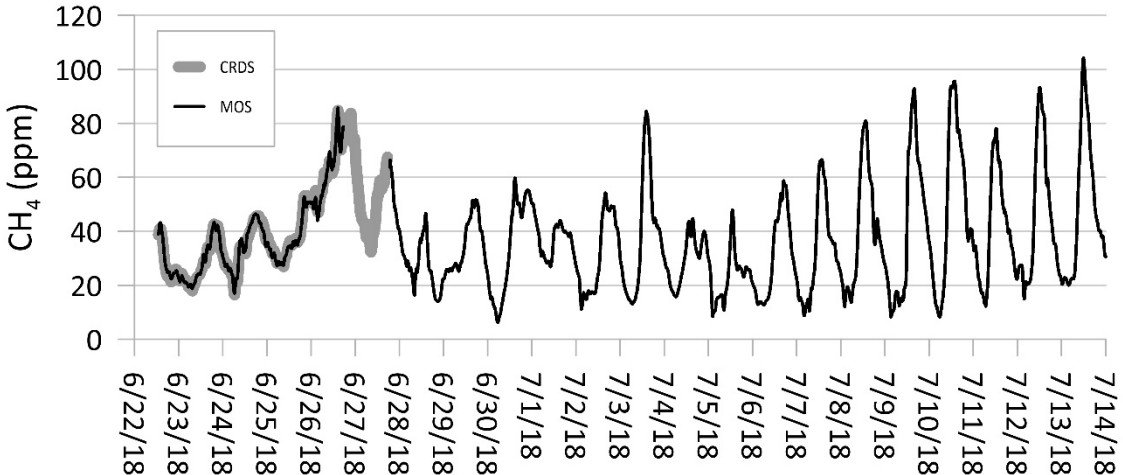


Figure 8: Smoothed time series measurements of CH$_4$ at the Greenland Ice Sheet using both the cavity ring-down spectroscopy (CRDS) reference monitor and the metal oxide sensor (MOS).



**Tables**

Table 1. Statistics for the calculated differences between the smoothed MOS and smoothed CDRS data series in both the non-turbulent time period and full field calibration period. The unit for error and difference values is ppm.

| Statistics:  $MOS_{smooth}$ - $CRDS_{smooth}$ | Non-Turbulent | Full series |
|---|---|---|
| Mean bias error | 0.09 | -0.05 |
| Root mean square error | 1.35 | 1.69 |
| Maximum negative difference | -3.96 | -11.83 |
| Maximum positive difference | 5.04 | 5.91 |
| Observations | 28501 | 37140 |
