# Peer review of "Continuous methane concentration measurements at the Greenland Ice Sheet-atmosphere interface using a low-cost low-power metal oxide sensor system"

_Atmospheric Measurement Techniques, 2019_

## Referee Comment (RC1) · Anonymous Referee #1 · 16 Jan 2020

General comments

This paper presents laboratory and field calibrations of a low-cost MOS for the measurement of methane concentrations in air. Whilst laser spectroscopy is currently the state-of-the-art solution for high-precision measurements of trace gases such as methane, this technology is expensive and ill-suited to remote, hostile environments such as the Greenland Ice Sheet, which is the study site of this paper. There is a great need to develop low-cost, low-power, rugged sensors capable of operating autonomously in remote locations and this is particularly critical for Arctic ecosystems

where the effects of climate change on greenhouse gas emissions are believed to be much larger than at lower latitudes. This paper compares a low-cost MOS with a state-of-the-art Picarro cavity ringdown spectrometer (CRDS), using the latter as a benchmark, and demonstrates the suitability of the prototype for real-time, in situ measurements. The proof-of-concept study is well-designed and generally adequately documented, and the subject matter is a good match for the scope of the journal. The technology is interesting and I hope that it will be developed further. This manuscript should be considered for publication provided that all the comments listed below are addressed.

Specific comments

The quality of English is acceptable but efforts should be made to shorten sentences throughout the manuscript.

Line 61: the MOS would not directly inform on climatic feedbacks. Please shorten the sentence to "... sensing element for future studies into CH4 emissions from the subglacial domain under the Greenland Ice Sheet."

Line 80: what is an open-ended enclosure?

Lines 85-91: Where/how was the air fed to the MOS sampled from? Through a 50 m tube, independently of the CRDS? If so, the sampling rate, and hence flushing rate of the MOS enclosure, would have been > 3 times that of the CRDS. My interpretation of this is that the autonomous setup would have been different from the calibration one and you would no longer compare like for like (direct comparison?). Please explain.

Line 131: also refers to comment above. "Parallel measurements..."; the setup is still unclear to me. Did you use separate sampling lines for the CRDS and the MOS?

Line 147: I don't understand why 0.042 is more complicated than 0.05. What uncertainty does rounding up (why not round down to 0.04 which is nearest?) add?

Line 165: a graph illustrating the differences in model parameters would be useful.

How significant are the differences between lab and field calibrations? In line 115 (lab calibration) you mentioned that the temperature was kept constant at around 22 °C. Was there a temperature effect in the field calibrations? Please, comment.

Lines 187-189: I do not understand the relevance of discussing the response time of a similar MOS, unless by similar you mean same model, different unit. Furthermore, the response time range (1-30 minutes) is massive compared to the CRDS (< 1 Hz). Considering the large differences in response times, you would have to take into consideration the temporal buffering introduced by the pumping rate, particularly where the Picarro is concerned ($\sim$ 47 seconds to flush the 50 m sampling tube @ 800 mL/min).

Lines 220-222: Re. filtering out the fluctuations attributed to micro-turbulence/ dilution of cavity air by influx of ambient air. If the purpose of the exercise is to study the emissions of CH4 from the cavity, then filtering out such perturbations is justified. However, this paper is concerned with a field assessment of a MOS sensor, and in this context, characterising the response of the 2 sensors to these perturbations is of great interest. This ties in with the comment above (response time and temporal buffering). Looking at Fig. 7a, the outliers in the turbulent period are further from the smoothed line for the CRDS than for the MOS. This might be an effect of the faster response time of the CRDS. It would be interesting to choose a longer averaging time (>= sample line flush rate + sensor response time) and plot the time series of Fig. 7a and b again. I would like to see this analysed and discussed rather than just smooth it out.

Line 284: "very close"; please quantify this statement.

Lines 295-297: Please tone down this statement. Your study evaluates a low-cost sensor for the measurement of CH4 in a hostile environment with the potential to lead to a better understanding and quantification of CH4 emissions from GrIS and similar locations.

Technical comments

Line 57: ". . . and in sensor network grids." This might require clarification.

Line 57: change "we have in situ tested. . ." to "we have tested in situ. . .".

Line 67: ". . .southern flank. . ." of what?. Terminus does not seem to be the right term.

Line 74-75: this sentence is clumsy and needs re-structuring. Suggestion "Humidity and temperature of the subglacial air were measured every 10 s using a combined sensor (. . .) mounted at the tip of the aluminium pole inserted into the cave. The data were recorded using. . ."

Section 2.2: could you specify whether the MOS setup was built by your lab?

Line 99: "electrical circuit converts" not convert.

Line 100: "were powered" not was.

Line 127: "are inversely" not is.

Lines 132-136: long sentence, difficult to read. Split into 2 parts.

Line 168: "which has been reported to scale linearly. . ."?

Line 177: "at the margin of the. . ." not if.

Line 211: "Measurements. . . show. . ." not shows.

Line 268: ". . . while being undetected. . ."

Line 284: "departs" means leaves. Use a more appropriate verb.

Line 313: ". . . which could significantly improve. . ."

Lines 314-317: this sentence is too long. Please divide it into two.

Line 325: Remove "very clean" unless you can substantiate its meaning.

Fig. 7: please indicate the temporal resolution of each plot.

Fig. 8: as in Fig. 7, what is the time step?

[Figure]

---

## Referee Comment (RC2) · Anonymous Referee #2 · 29 Feb 2020

Review of manuscript amt-2019-468

General aspects:

This is a well-written and interesting study showing how low cost metal oxide semiconductor sensors (MOS) for methane (CH4) can be used to follow CH4 mixing ratios over time in Greenland glacier ice caves. Results convincingly indicate that MOS sensors can perform very well and this is promising for easier and less costly monitoring under such conditions (very stable temperature and relative humidity). These tests are

important and I congratulate the authors for their careful and interesting work.

The authors are asked to consider the specific comments below in the revision of the manuscript.

Specific comments (numbers refer to line numbers):

15. Please define CRDS in abstract. Some readers may not be familiar with cavity ring-down spectrometry.

19-20: What was MBE selected instead of MAE or RMSE? With MBE, positive and negative bias cancel out which is not desirable. Please consider using RMSE or MAE instead.

97-98. Is it really correct that the conductivity increase with gas concentration as indicated here? Does not the output voltage increase with CH4 mixing ratio due to increasing resistance at higher CH4 levels, which would mean reduced conductivity?

120. Eq.1: What is R0 in Figure 3? Is it equivalent to Rs? If so, please consider using consistent notation in both text, figures and tables.

139-148: Please here explain why the smoothing was needed. An explanation is given later in the text, but it would be good for understanding to provide the explanation here.

155-160 and elsewhere. At less stable conditions than in the ice cave studied here, it would be challenging to have zero gas and sample gas with the same water concentrations. Hence, correction to humidity seems needed. Please see doi.org/10.5194/bg-2019-499 for detailed analyses of ways to correct for humidity and temperature to derive more generally applicable calibration curves.

163-165- Unclear how the rather poor fit in Figure 6 between MOS and CRDS could be translated into the very close fit in Figure 7. Please clarify this in the manuscript.

163-174. Could the deviation between the lab and the field be due to any other factors?

239-240. This statement gives the impression that the MOS are accurate to 10 ppb level. Is this really correct? This is orders of magnitude better than others have found. The mean bias error is risky to use because negative and positive errors cancel out. Please consider using RMSE as indicator of MOS performance.

243-254. Would not field calibration also be an option as done here and suggested in doi.org/10.5194/bg-2019-499? Given the low temperature - what was the absolute humidity which is what influence sensors more than RH?

305-307. Some of this is addressed in doi.org/10.5194/bg-2019-499 which could be worth citing.

323-324. Please see previous comments regarding MBE vs RMSE.

484-485. Please clarify what FIgure 6 shows in relation to Figures 7 and 8. The offset between the sensor and CRDS data are much greater in Figure 6 than in Figure 7 and 8. Figure 6 looks more like what could be expected from theses sensors, while the fit versus the CDRS in Figure 7 and 8 is extremely close (looks fantastic and almost too good to be true, and it is hard to undestand how the calbration equatinos provided could correct all the offset in Figure). Hence, clarifying the differences between Figure 6 vs 7 and 8 seem very important for fully understanding the study and proper sensor use.

490-496. Legend of Figure 7 has many abbreviations. Please consider to define or spell them out to make it easier to understand the figure independently from the main text? Also it would be of great interest to readers to add humidity to the figure.

---

## Referee Comment (RC3) · Anonymous Referee #3 · 30 Mar 2020

In this paper, the authors studied the performance of a low-cost and low-power methane ($CH_4$) sensing system prototype based on a metal oxide sensor (MOS) sensitive to $CH_4$. The sensor was tested in a natural $CH_4$ emitting environment at the Greenland Ice sheet (GrIS). The primary scientific importance of the study is that it provides a clear example on how the application of low cost technology can enhance our future understanding on the climatic feedbacks from the cryosphere to the atmosphere.

The present study fits within the aim of this journal and the results are promising and

interesting for future applications of low cost sensors.

The reviewer think that the paper can be published for open discussion and a main lack has been observed: - Low costs sensors from past studies show a 'drift' of the sensors response over the time. The authors do not cite this problem and neither they have tested it because a short experiment has been performed. This should be underline and future studies should include long term comparison between reference instrument and low cost sensor kit. The correction for the drift of the sensor will increase the final uncertainty related to the measurement and will also increase the cost of the field campaign because of the need of in situ continuous calibrations. The reviewer suggests to perform a study on the sensor drift over the months.

———————————————————

---

## Author Comment (AC1) · 22 Apr 2020

Dear Anonymous Reviewer #1. Thank you very much for your help in improving the manuscript. Please find our detailed point-by-point to your constructive criticism of our manuscript in the included file "Combined point-by-point responses to reviewer's comments"

"General comments This paper presents laboratory and field calibrations of a low-cost MOS for the measurement of methane concentrations in air. Whilst laser spectroscopy

is currently the state-of-the-art solution for high-precision measurements of trace gases such as methane, this technology is expensive and ill-suited to remote, hostile environments such as the Greenland Ice Sheet, which is the study site of this paper. There is a great need to develop low-cost, low-power, rugged sensors capable of operating autonomously in remote locations and this is particularly critical for Arctic ecosystems where the effects of climate change on greenhouse gas emissions are believed to be much larger than at lower latitudes. This paper compares a low-cost MOS with a state-of-the-art Picarro cavity ringdown spectrometer (CRDS), using the latter as a benchmark, and demonstrates the suitability of the prototype for real-time, in situ measurements. The proof-of-concept study is well-designed and generally adequately documented, and the subject matter is a good match for the scope of the journal. The technology is interesting and I hope that it will be developed further. This manuscript should be considered for publication provided that all the comments listed below are addressed."

⇒ Reply 1: We appreciate the constructive criticism by Reviewer #1, #2 and #3. We have prepared a point-by-point response to each of the raised issues below, and incorporated appropriate changes to the manuscript, accordingly:

"Specific comments: The quality of English is acceptable but efforts should be made to shorten sentences throughout the manuscript."

⇒ Reply 2: Ok.

"Line 61: the MOS would not directly inform on climatic feedbacks. Please shorten the sentence to ": : : sensing element for future studies into CH4 emissions from the subglacial domain under the Greenland Ice Sheet.""

⇒ Reply 3: Suggestion has been followed. The revised sentence is: "This was done to assess the MOS's potential for serving as a sensing element for future studies CH4 emissions from the subglacial domain under the Greenland Ice Sheet. "

"Line 80: what is an open-ended enclosure? (A)"

"Lines 85-91: Where/how was the air fed to the MOS sampled from? Through a 50 m tube, independently of the CRDS? If so, the sampling rate, and hence flushing rate of the MOS enclosure, would have been > 3 times that of the CRDS. My interpretation of this is that the autonomous setup would have been different from the calibration one and you would no longer compare like for like (direct comparison?). Please explain. (B)"

"Line 131: also refers to comment above. "Parallel measurements: : :"; the setup is still unclear to me. Did you use separate sampling lines for the CRDS and the MOS? (C)"

⇒ Reply 4: A combined reply has been prepared for the three above reviewer comments (A,B,C)

Two different configurations were used depending on the measurement period:

1) Field calibration period where parallel measurements were done with the CRDS and MOS connected in series. In this configuration, a 50 meter plastic tube connected the subglacial sampling point to the inlet of CRDS. Here, the sample gas passed through the internal pump of the CRDS to the measurement cell before exiting the outlet port of the CRDS. The outlet port was connected via 1 meter tube to enclosure where the MOS was placed. 2) Autonomous measuring period where the CRDS was replaced by a small 12 volt diaphragm pump (inlet of pump connected to the sampling point and outlet of pump connected to bottom of enclosure).

In order to make this more clear as well as to accommodate the general advise of shortening sentences. the 2nd and 3rd paragraph of section 2.1 has been revised to the following:

"Real-time reference concentration measurements of $CH_4$, carbon dioxide ($CO_2$) and water vapor ($H_2O$) was obtained using a CRDS (Ultraportable Greenhouse Gas Analyzer, Los Gatos Research, USA). The inlet port of the CRDS was connected to the

subglacial sampling point via a sampling tube (50 m length, inner diameter of 4 mm and total volume of 630 mL) which was zip-tied to the aluminium pole. Flow of sample gas from the subglacial sampling point to the measurement cell in the CRDS was obtained via the analyzer's internal diaphragm pump (800 mL min-1). The outlet port of the CRDS was connected in series via a 1 m plastic tube to a metal can enclosure (400 mL), where the lid had been removed (Fig. 2b). The prototype CH4 sensing system (MOS) was placed in the metal enclosure, where the short serial tube connector ensured a rapid flushing of the headspace in which the CH4 measurements with the MOS were made. Due to the non-destructive sampling principle of the CRDS and the rapid flushing of the headspace volume in the enclosure with the MOS system (2 times per minute), the concentration of CH4 is estimated to be virtually identical at the same time step for the MOS and the CRDS during the entire field calibration period (22nd to 26th July 2018).

Following the field calibration test of approximately 100 h, the MOS system was left in the field as an autonomous monitoring system. For this autonomous measurement period, the CRDS was replaced by a 12 volt diaphragm pump (Thomas pumps, 1410VD DC) with a constant air-flow of approximately 3 L min-1 attached to the common sample tube with similar connection of the pump inlet and outlet as the CRDS ports. During this period the MOS system was powered by 12V LiFePO4 batteries connected to solar panels and a voltage regulator, placed in a water-proof case and buried under a pile of rocks to minimize the impact of sunlight induced temperature variations of the sensor system. "

Figure caption of Fig.2 has also been updated for improved clarity.

Also, the wording "Parallel measurements" has been changed throughout the manuscript to "simultaneous measurement" to avoid the potential ambiguity of whether the CRDS and MOS were connected in series using a common sample tube (as were the case) or in parallel using different sample tubes (which were not the case).

[Figure]

"Line 147: I don't understand why 0.042 is more complicated than 0.05. What uncertainty does rounding up (why not round down to 0.04 which is nearest?) add?"

⇒ Reply 5: In a sense, the reviewer could be right that it defies its own purpose to do an optimization for a best values, and then round it up afterwards. We have revised the data smoothing with 0.42 for both dataseries, and updated the figures accordingly.

"Line 165: a graph illustrating the differences in model parameters would be useful. How significant are the differences between lab and field calibrations? In line 115 (lab calibration) you mentioned that the temperature was kept constant at around 22 _C. Was there a temperature effect in the field calibrations? Please, comment."

⇒ Reply 6: The environmental conditions between the controlled atmosphere of the laboratory and the uncontrolled field conditions in Greenland are of course significantly different, which is one of the reasons why field calibration of the MOS seems necessary, unless we work out a better way to do a generic standard calibration. During the field measurements used for the calculation of the R0*, air with a CRDS confirmed CH4 concentration of the atmospheric background (approx. 1.9 ppm CH4) was sampled within 10 meters of the ice margin where meltwater and CH4 emission was absent. The exact temperature and relative humidity of this air mass is unknown, but likely within the range between 1-4 Co and above 90 % RH. The text in section "2.4 Field calibration of the MOS" has been revised to include this information.

"Lines 187-189: I do not understand the relevance of discussing the response time of a similar MOS, unless by similar you mean same model, different unit. Furthermore, the response time range (1-30 minutes) is massive compared to the CRDS (< 1 Hz). Considering the large differences in response times, you would have to take into consideration the temporal buffering introduced by the pumping rate, particularly where the Picaro is concerned (_ 47 seconds to flush the 50 m sampling tube @ 800 mL/min)."

⇒ Reply 7: Since we cannot be absolutely sure that the model used in the reference is identical to the TGS2611 used in this study, we have followed the advice to remove

this part of the discussion.

"Lines 220-222: Re. filtering out the fluctuations attributed to micro-turbulence/ dilution of cavity air by influx of ambient air. If the purpose of the exercise is to study the emissions of CH4 from the cavity, then filtering out such perturbations is justified.

However, this paper is concerned with a field assessment of a MOS sensor, and in this context, characterizing the response of the 2 sensors to these perturbations is of great interest. This ties in with the comment above (response time and temporal buffering). Looking at Fig. 7a, the outliers in the turbulent period are further from the smoothed line for the CRDS than for the MOS. This might be an effect of the faster response time of the CRDS. It would be interesting to choose a longer averaging time (>= sample line flush rate + sensor response time) and plot the time series of Fig. 7a and b again. I would like to see this analyzed and discussed rather than just smooth it out."

⇒ Reply 8: We understand the point raised by the reviewer, and agree that a better understanding of especially the response of the MOS to fluctuation conditions would be great. In the current dataset, we unfortunately have no data to actually quantify the amount and rate of dilution by microturbulens to support such an analysis. Since the ultimate aim of this study is to develop a low-cost low-power system specifically designed to study subglacial CH4 emissions, we feel more comfortable with proceeding the data smoothing as presented, and thereby avoiding the risk of over-analyzing a dataset with respect to parameters which are uncontrolled.

"Line 284: "very close"; please quantify this statement."

⇒ Reply 9: See revised sentence under reply 24.

"Lines 295-297: Please tone down this statement. Your study evaluates a low-cost sensor for the measurement of CH4 in a hostile environment with the potential to lead to a better understanding and quantification of CH4 emissions from GrIS and similar locations."

⇒ Reply 10: OK. The sentence has been removed in the revised MS.

"Technical comments paper Line 57: ": : : and in sensor network grids." This might require clarification."

⇒ Reply 11: text has been changed to "…sensor networks."

"Line 57: change "we have in situ tested: : :" to "we have tested in situ: : :"."

⇒ Reply 12: Corrected.

"Line 67: ": : :southern flank: : :" of what?. Terminus does not seem to be the right term."

⇒ Reply 13: "… at the terminus…" has been removed in the revised MS.

"Line 74-75: this sentence is clumsy and needs re-structuring. Suggestion "Humidity and temperature of the subglacial air were measured every 10 s using a combined sensor (: : :) mounted at the tip of the aluminium pole inserted into the cave. The data were recorded using: : :""

⇒ Reply 14: Good suggestion. Text has been replaced as suggested.

"Section 2.2: could you specify whether the MOS setup was built by your lab?"

⇒ Reply 15: The following has been added: "The final prototype was assembled in the laboratory at Aarhus University."

"Line 99: "electrical circuit converts" not convert."

⇒ Reply 16: Corrected.

"Line 100: "were powered" not was."

⇒ Reply 17: Corrected. "Line 127: "are inversely" not is."

⇒ Reply 18: Corrected.

[Figure]

"Lines 132-136: long sentence, difficult to read. Split into 2 parts."

⇒ Reply 19: Sentence now reads: "Access to a controlled and humidified zero gas was not available in the field. Instead the atmospheric background concentration of $CH_4$ of the air (approximately 1.9 ppm) close to the ice sheet was used to calculate the average ambient sensor resistance (R0*) using Eq. 1. The output value of the MOS under these conditions was then used to establish the resistance ratio (RS/R0*) vs. $CH_4$ concentration field calibration function for the MOS (Fig. 6)."

"Line 168: "which has been reported to scale linearly: : :"?"

⇒ Reply 20: Sorry, incomplete sentence. The revised wording is:

The reason for this difference is unknown, but a possible explanation could be the potential difference in input heater voltage for the MOS sensor (i.e. pin 1 and 4 in Fig. 1), since variations in the input heater voltage have been reported to affect the $CH_4$ concentration measurements (van den Bossche et al., 2017).

"Line 177: "at the margin of the: : :" not if."

⇒ Reply 21: Corrected.

"Line 211: "Measurements: : : show: : :" not shows."

⇒ Reply 22: Corrected.

"Line 268: ": : : while being undetected: : :""

⇒ Reply 23: Corrected.

"Line 284: "departs" means leaves. Use a more appropriate verb."

⇒ Reply 24: The sentence has been reformulated: "CRDSsmooth data for period 2 fills the data gap between the MOS measurement of period 1 and 3, where the start concentration data of the MOSsmooth concentration data are similar to the concentration level where the CRDSsmooth measurements end

"Line 313: ": : : which could significantly improve: : :"""

⇒ Reply 25: Corrected.

"Lines 314-317: this sentence is too long. Please divide it into two."

⇒ Reply 26: Corrected.

"Line 325: Remove "very clean" unless you can substantiate its meaning."

⇒ Reply 27: Corrected.

"Fig. 7: please indicate the temporal resolution of each plot."

⇒ Reply 28: Temporal resolution is 10 seconds. Info has been added to the figure caption.

"Fig. 8: as in Fig. 7, what is the time step?"

⇒ Reply 29: Temporal resolution is 10 seconds. Info has been added to the figure caption.

Please also note the supplement to this comment:
https://www.atmos-meas-tech-discuss.net/amt-2019-468/amt-2019-468-AC1-supplement.pdf

---

## Author Comment (AC2) · 22 Apr 2020

Dear Anonymous Reviewer #2. Thank you very much for your help in improving the manuscript. Please find our detailed point-by-point to your constructive criticism of our manuscript in the included file "Combined point-by-point responses to reviewer's comments". Additional figure material for reply 32 and 43 is available in the supplement pdf file.

"Anonymous Referee #2 Review of

manuscript amt-2019-468 General aspects:

This is a well-written and interesting study showing how low cost metal oxide semiconductor sensors (MOS) for methane (CH4) can be used to follow CH4 mixing ratios over time in Greenland glacier ice caves. Results convincingly indicate that MOS sensors can perform very well and this is promising for easier and less costly monitoring under such conditions (very stable temperature and relative humidity). These tests are important and I congratulate the authors for their careful and interesting work. The authors are asked to consider the specific comments below in the revision of the manuscript.

Specific comments (numbers refer to line numbers): 15. Please define CRDS in abstract. Some readers may not be familiar with cavity ring-down spectrometry."

⇒ Reply 30: Corrected.

"19-20: What was MBE selected instead of MAE or RMSE? With MBE, positive and negative bias cancel out which is not desirable. Please consider using RMSE or MAE instead."

⇒ Reply 31: This could be an area prone to confusion, and we agree to resolved this in the revised manuscript by including both the mean bias error (MBE), mean absolute error (MAE) and root mean square error (RMSE) in table 1, and replace the MBE in the abstract with the RMSE value.

"97-98. Is it really correct that the conductivity increase with gas concentration as indicated here? Does not the output voltage increase with CH4 mixing ratio due to increasing resistance at higher CH4 levels, which would mean reduced conductivity?"

⇒ Reply 32: In the description of the circuit (https://www.figaro.co.jp/en/product/docs/tgs2611-e00_product%20information%28en%29_rev00.pdf), the output voltage of the sensor (VRL) is measured across a voltage divider with a fixed load resistor. The resistance of the variable resistor (Rs) (i.e. the sensing element itself) can be calculated based on the measured output voltage (VRL) according to the

following formula:

[SEE FIGURE IN SUPPLEMENT PDF FILE]

As the reviewer correctly points out, the output voltage (VRL) increased with increasing CH4 concentration. In our setup, we used a load resistor of 10 kOhm and a circuit voltage (VC) of 5 volt.

As argument for why we believe the description in the MS is correct consider the following simplified example: With an output value of say 1.5 volt (arbitrary lower CH4 concentration), the sensor resistance would be Rs: (5V/1.5V − 1) * 10kOhm = 23 kOhm, where a higher output value of say 2.5 volt (arbitrary higher CH4 concentration) would give an sensor resistance of 10 kOhm. In this way, higher CH4 concentrations produce lower sensor resistance or higher sensor conductivity.

"120. Eq.1: What is R0 in Figure 3? Is it equivalent to Rs? If so, please consider using consistent notation in both text, figures and tables."

⇒ Reply 33: R0 is equal to the sensor resistance Rs under the defined zero gas conditions, to which all other measurements are normalized to (i.e. to calculate the Rs/R0 ratio). Figure 3 only shows the simplified schematic of the electrical circuit, which does not include a notation of the calculated value of sensor resistance at zero gas conditions.

"139-148: Please here explain why the smoothing was needed. An explanation is given later in the text, but it would be good for understanding to provide the explanation here."

⇒ Reply 34: We have revised the sentence to be: "In order to compensate for potential effects of micro-turbulent mixing of subglacial air with atmospheric air (see also section 3.3), the measured raw time series data from the MOS were smoothed using simple exponential smoothing according to Eq. (2):.……."

"155-160 and elsewhere. At less stable conditions than in the ice cave studied here, it would be challenging to have zero gas and sample gas with the same water concentrations. Hence, correction to humidity seems needed. Please see doi.org/10.5194/bg-2019-499 for detailed analyses of ways to correct for humidity and temperature to derive more generally applicable calibration curves."

⇒ Reply 35: Thank you for providing the reference to an exciting and very relevant study, which has been published for discussion at a similar point in time as this study. We will include the reference to the list of references and include it in the revised discussion of the potential effect of water vapor and temperature on the sensor measurements.

"163-165- Unclear how the rather poor fit in Figure 6 between MOS and CRDS could be translated into the very close fit in Figure 7. Please clarify this in the manuscript."

⇒ Reply 36: We are not sure we understand what is meant by poor fit in figure 6 ($R^2$ = 0.98 / p-value = 0.001). However, there is a notable spread in some of the values of under higher concentration levels under more turbulent environmental conditions at the measurement point. However, it is difficult to visualize all of the 37.140 data points in only single figure without a great number of similar data points are visually stacked on top of each other, thereby not being visible in the figure. In this way, the apparent higher degree of spread in the upper CH4 levels are a product of the turbulent data (which are non-dominating for the statistical model) having a visual prevalence over the non-turbulent data points which dominate the statistical model.

The following text has been added to the revised MS in section3.2. to clarify the issue:

"A total of 37,140 data points are included in the regression model for converting the RS/RO* ratios to CH4 concentrations. Inclusion of data points from the micro-turbulent periods produces a noisy visualization of the calibrations data at higher CH4 concentration levels (Fig. 6). However, this apparent noise is primarily a visual artefact that does not have significance for the underlying calibration statistics, which shows excellent statistical agreement between the independent and dependent variables ($R^2$ = 0.98; p-value: 0.001)."

"163-174. Could the deviation between the lab and the field be due to any other factors?"

⇒ Reply 37: It is possible, yes, but further experimental and calibration work would be needed to bring more certainty to this issue.

"239-240. This statement gives the impression that the MOS are accurate to 10 ppb level. Is this really correct? This is orders of magnitude better than others have found. The mean bias error is risky to use because negative and positive errors cancel out. Please consider using RMSE as indicator of MOS performance."

⇒ Reply 38: We agree to use the RMSE as suggested and have updated the sentence accordingly.

"243-254. Would not field calibration also be an option as done here and suggested in doi.org/10.5194/bg-2019-499? Given the low temperature - what was the absolute humidity which is what influence sensors more than RH?"

⇒ Reply 39: We agree. The original wording was chosen to reflect what has previously been done. The sentence is updated with the inclusion of the suggested reference. The saturated water vapor or absolute humidity in the thermally buffered measurement environment around 0 degrees C is approximately 5 g/m3 (see http://hyperphysics.phy-astr.gsu.edu/hbase/Kinetic/watvap.html).

"305-307. Some of this is addressed in doi.org/10.5194/bg-2019-499 which could be worth citing."

⇒ Reply 40: We agree. Citations have been added.

"323-324. Please see previous comments regarding MBE vs RMSE."

⇒ Reply 41: We have rephrased to focus on the RMSE

"484-485. Please clarify what FIgure 6 shows in relation to Figures 7 and 8. The offset between the sensor and CRDS data are much greater in Figure 6 than in Figure 7 and

8. Figure 6 looks more like what could be expected from theses sensors, while the fit versus the CDRS in Figure 7 and 8 is extremely close (looks fantastic and almost too good to be true, and it is hard to understand how the calibration equations provided could correct all the offset in Figure). Hence, clarifying the differences between Figure 6 vs 7 and 8 seem very important for fully understanding the study and proper sensor use."

⇒ Reply 42: Please see reply 36 for the issue of data fit (Figure 6) and reply 44 for precision issue. In short, our results are in line with the finding of also https://doi.org/10.5194/amt-2019-402, where the authors conclude ". . . that the TGS 2600 sensor can provide data of research-grade quality if it is adequately calibrated and placed in a suitable environment where cross-sensitivities to gases other than CH4 is of no concern."

"490-496. Legend of Figure 7 has many abbreviations. Please consider to define or spell them out to make it easier to understand the figure independently from the main text? Also it would be of great interest to readers to add humidity to the figure."

⇒ Reply 43: Definitions have been added to captions for Figure 7.

With respect to the relative humidity. The resolution of the used RH sensor (S-THB-M008 from Onset;) is stated by the manufacturer to be 0.1 % RH (https://www.onsetcomp.com/products/sensors/s-thb-m008/). From our measurements, the results show a flat line of 100% relative humidity over the entire measurement period. We therefore chose to only describe this result by means of text (line 251 in original MS) and not show a flat in the figure.

[SEE FIGURE IN SUPPLEMENT PDF FILE]

To accommodate the comment from reviewer 2, we have expanded the description of the relative humidity measurements in the text, to better illustrate that RH was extremely stable in the measurement environment (resolution of sensor has been added

in section 2.1 and text has been revised in section 3.4).

"Measurements from the air-filled cavity under the GrIS document a very stable sampling environment with a relative humidity throughout the sampling period showing consistent readings of 100 % RH (data not shown) and only minor air temperature variations between approximately 0.05 oC during the night and 0.25 oC during mid-day (Fig. 7d)."

Please also note the supplement to this comment:
https://www.atmos-meas-tech-discuss.net/amt-2019-468/amt-2019-468-AC2-supplement.pdf

**Supplement:**

**Point-by-point response to comments by Anonymous Referees #1, #2 and #3 for manuscript "amt-2019-468".**

Anonymous Referee #1

"General comments
This paper presents laboratory and field calibrations of a low-cost MOS for the measurement of methane concentrations in air. Whilst laser spectroscopy is currently the state-of-the-art solution for high-precision measurements of trace gases such as methane, this technology is expensive and ill-suited to remote, hostile environments such as the Greenland Ice Sheet, which is the study site of this paper. There is a great need to develop low-cost, low-power, rugged sensors capable of operating autonomously in remote locations and this is particularly critical for Arctic ecosystems where the effects of climate change on greenhouse gas emissions are believed to
be much larger than at lower latitudes. This paper compares a low-cost MOS with a state-of-the-art Picarro cavity ringdown spectrometer (CRDS), using the latter as a benchmark, and demonstrates the suitability of the prototype for real-time, in situ measurements. The proof-of-concept study is well-designed and generally adequately documented, and the subject matter is a good match for the scope of the journal. The technology is interesting and I hope that it will be developed further. This manuscript should be considered for publication provided that all the comments listed below are addressed."

➔ Reply 1: We appreciate the constructive criticism by Reviewer #1, #2 and #3. We have prepared a point-by-point response to each of the raised issues below, and incorporated appropriate changes to the manuscript, accordingly:

"Specific comments:
The quality of English is acceptable but efforts should be made to shorten sentences throughout the manuscript."

➔ Reply 2: Ok.

"Line 61: the MOS would not directly inform on climatic feedbacks. Please shorten the sentence to ": : : sensing element for future studies into CH4 emissions from the subglacial domain under the Greenland Ice Sheet.""

➔ Reply 3: Suggestion has been followed. The revised sentence is: "*This was done to assess the MOS's potential for serving as a sensing element for future studies $CH_4$ emissions from the subglacial domain under the Greenland Ice Sheet.* "

"Line 80: what is an open-ended enclosure? (**A**)"

"Lines 85-91: Where/how was the air fed to the MOS sampled from? Through a 50 m tube, independently of the CRDS? If so, the sampling rate, and hence flushing rate of the MOS enclosure, would have been > 3 times that of the CRDS. My interpretation of this is that the autonomous setup would have been different from the calibration one and you would no longer compare like for like (direct comparison?). Please explain. (**B**)"

"Line 131: also refers to comment above. "Parallel measurements: : :"; the setup is still

unclear to me. Did you use separate sampling lines for the CRDS and the MOS? (**C**)"

➔ Reply 4: A combined reply has been prepared for the three above reviewer comments (**A,B,C**)

Two different configurations were used depending on the measurement period:

1) Field calibration period where parallel measurements were done with the CRDS and MOS **connected in series**. In this configuration, a 50 meter plastic tube connected the subglacial sampling point to the inlet of CRDS. Here, the sample gas passed through the internal pump of the CRDS to the measurement cell before exiting the outlet port of the CRDS. The outlet port was connected via 1 meter tube to enclosure where the MOS was placed.
2) Autonomous measuring period where the CRDS was replaced by a small 12 volt diaphragm pump (inlet of pump connected to the sampling point and outlet of pump connected to bottom of enclosure).

In order to make this more clear as well as to accommodate the general advise of shortening sentences. the $2^{nd}$ and $3^{rd}$ paragraph of section 2.1 has been revised to the following:

"*Real-time reference concentration measurements of $CH_4$, carbon dioxide ($CO_2$) and water vapor ($H_2O$) was obtained using a CRDS (Ultraportable Greenhouse Gas Analyzer, Los Gatos Research, USA). The inlet port of the CRDS was connected to the subglacial sampling point via a sampling tube (50 m length, inner diameter of 4 mm and total volume of 630 mL) which was zip-tied to the aluminium pole. Flow of sample gas from the subglacial sampling point to the measurement cell in the CRDS was obtained via the analyzer's internal diaphragm pump (800 mL min$^{-1}$). The outlet port of the CRDS was connected in series via a 1 m plastic tube to a metal can enclosure (400 mL), where the lid had been removed (Fig. 2b). The prototype $CH_4$ sensing system (MOS) was placed in the metal enclosure, where the short serial tube connector ensured a rapid flushing of the headspace in which the $CH_4$ measurements with the MOS were made. Due to the non-destructive sampling principle of the CRDS and the rapid flushing of the headspace volume in the enclosure with the MOS system (2 times per minute), the concentration of $CH_4$ is estimated to be virtually identical at the same time step for the MOS and the CRDS during the entire field calibration period ($22^{nd}$ to $26^{th}$ July 2018).*

*Following the field calibration test of approximately 100 h, the MOS system was left in the field as an autonomous monitoring system. For this autonomous measurement period, the CRDS was replaced by a 12 volt diaphragm pump (Thomas pumps, 1410VD DC) with a constant air-flow of approximately 3 L min$^{-1}$ attached to the common sample tube with similar connection of the pump inlet and outlet as the CRDS ports. During this period the MOS system was powered by 12V LiFePO$_4$ batteries connected to solar panels and a voltage regulator, placed in a water-proof case and buried under a pile of rocks to minimize the impact of sunlight induced temperature variations of the sensor system.*"

Figure caption of Fig.2 has also been updated for improved clarity.

Also, the wording "***Parallel measurements***" has been changed throughout the manuscript to "***simultaneous measurement***" to avoid the potential ambiguity of whether the CRDS and MOS were connected in series using a common sample tube (as were the case) or in parallel using different sample tubes (which were not the case).

"Line 147: I don't understand why 0.042 is more complicated than 0.05. What uncertainty does rounding up (why not round down to 0.04 which is nearest?) add?"

➔ Reply 5: In a sense, the reviewer could be right that it defies its own purpose to do an optimization for a best values, and then round it up afterwards. We have revised the data smoothing with 0.42 for both dataseries, and updated the figures accordingly.

"Line 165: a graph illustrating the differences in model parameters would be useful. How significant are the differences between lab and field calibrations? In line 115 (lab calibration) you mentioned that the temperature was kept constant at around 22 _C. Was there a temperature effect in the field calibrations? Please, comment."

➔ Reply 6: The environmental conditions between the controlled atmosphere of the laboratory and the uncontrolled field conditions in Greenland are of course significantly different, which is one of the reasons why field calibration of the MOS seems necessary, unless we work out a better way to do a generic standard calibration. During the field measurements used for the calculation of the $R_0$*, air with a CRDS confirmed $CH_4$ concentration of the atmospheric background (approx. 1.9 ppm $CH_4$) was sampled within 10 meters of the ice margin where meltwater and CH4 emission was absent. The exact temperature and relative humidity of this air mass is unknown, but likely within the range between 1-4 $C^o$ and above 90 % RH. The text in section "2.4 Field calibration of the MOS" has been revised to include this information.

"Lines 187-189: I do not understand the relevance of discussing the response time of a similar MOS, unless by similar you mean same model, different unit. Furthermore, the response time range (1-30 minutes) is massive compared to the CRDS (< 1 Hz). Considering the large differences in response times, you would have to take into consideration the temporal buffering introduced by the pumping rate, particularly where the Picaro is concerned (_ 47 seconds to flush the 50 m sampling tube @ 800 mL/min)."

➔ Reply 7: Since we cannot be absolutely sure that the model used in the reference is identical to the TGS2611 used in this study, we have followed the advice to remove this part of the discussion.

"Lines 220-222: Re. filtering out the fluctuations attributed to micro-turbulence/ dilution of cavity air by influx of ambient air. If the purpose of the exercise is to study the emissions of CH4 from the cavity, then filtering out such perturbations is justified.

However, this paper is concerned with a field assessment of a MOS sensor, and in this context, characterizing the response of the 2 sensors to these perturbations is of great interest. This ties in with the comment above (response time and temporal buffering). Looking at Fig. 7a, the outliers in the turbulent period are further from the smoothed line for the CRDS than for the MOS. This might be an effect of the faster response time of the CRDS. It would be interesting to choose a longer averaging time (>= sample line flush rate + sensor response time) and plot the time series of Fig. 7a and b again. I would like to see this analyzed and discussed rather than just smooth it out."

➔ Reply 8: We understand the point raised by the reviewer, and agree that a better understanding of especially the response of the MOS to fluctuation conditions would be great. In the current dataset, we unfortunately have no data to actually quantify the amount and rate of dilution by microturbulens to

support such an analysis. Since the ultimate aim of this study is to develop a low-cost low-power system specifically designed to study subglacial CH4 emissions, we feel more comfortable with proceeding the data smoothing as presented, and thereby avoiding the risk of over-analyzing a dataset with respect to parameters which are uncontrolled.

"Line 284: "very close"; please quantify this statement."

➔ Reply 9: See revised sentence under reply 24.

"Lines 295-297: Please tone down this statement. Your study evaluates a low-cost sensor for the measurement of CH4 in a hostile environment with the potential to lead to a better understanding and quantification of CH4 emissions from GrIS and similar locations."

➔ Reply 10: OK. The sentence has been removed in the revised MS.

"Technical comments

Line 57: ": : : and in sensor network grids." This might require clarification."

➔ Reply 11: text has been changed to "…sensor networks."

"Line 57: change "we have in situ tested: : :" to "we have tested in situ: : :"."

➔ Reply 12: Corrected.

"Line 67: ": : :southern flank: : :" of what?. Terminus does not seem to be the right term."

➔ Reply 13: "… *at the terminus*…" has been removed in the revised MS.

"Line 74-75: this sentence is clumsy and needs re-structuring. Suggestion "Humidity and temperature of the subglacial air were measured every 10 s using a combined sensor (: : :) mounted at the tip of the aluminium pole inserted into the cave. The data were recorded using: : :""

➔ Reply 14: Good suggestion. Text has been replaced as suggested.

"Section 2.2: could you specify whether the MOS setup was built by your lab?"

➔ Reply 15: The following has been added: "The final prototype was assembled in the laboratory at Aarhus University."

"Line 99: "electrical circuit converts" not convert."

➔ Reply 16: Corrected.

"Line 100: "were powered" not was."

➔ Reply 17: Corrected.

"Line 127: "are inversely" not is."

➔ Reply 18: Corrected.

"Lines 132-136: long sentence, difficult to read. Split into 2 parts."

➔ Reply 19: Sentence now reads: "Access to a controlled and humidified zero gas was not available in the field. Instead the atmospheric background concentration of $CH_4$ of the air (approximately 1.9 ppm) close to the ice sheet was used to calculate the average ambient sensor resistance ($R_{0*}$) using Eq. 1. The output value of the MOS under these conditions was then used to establish the resistance ratio ($R_S/R_{0*}$) vs. $CH_4$ concentration field calibration function for the MOS (Fig. 6)."

"Line 168: "which has been reported to scale linearly: : :"?"

➔ Reply 20: Sorry, incomplete sentence. The revised wording is:

The reason for this difference is unknown, but a possible explanation could be the potential difference in input heater voltage for the MOS sensor (i.e. pin 1 and 4 in Fig. 1), since variations in the input heater voltage have been reported to affect the $CH_4$ concentration measurements (van den Bossche et al., 2017).

"Line 177: "at the margin of the: : :" not if."

➔ Reply 21: Corrected.

"Line 211: "Measurements: : : show: : :" not shows."

➔ Reply 22: Corrected.

"Line 268: ": : : while being undetected: : :""

➔ Reply 23: Corrected.

"Line 284: "departs" means leaves. Use a more appropriate verb."

➔ Reply 24: The sentence has been reformulated: "$CRDS_{smooth}$ data for period 2 fills the data gap between the MOS measurement of period 1 and 3, where the start concentration data of the $MOS_{smooth}$ concentration data are similar to the concentration level where the $CRDS_{smooth}$ measurements end

"Line 313: ": : : which could significantly improve: : :""

➔ Reply 25: Corrected.

"Lines 314-317: this sentence is too long. Please divide it into two."

➔ Reply 26: Corrected.

"Line 325: Remove "very clean" unless you can substantiate its meaning."

➔ Reply 27: Corrected.

"Fig. 7: please indicate the temporal resolution of each plot."

➔ Reply 28: Temporal resolution is 10 seconds. Info has been added to the figure caption.

"Fig. 8: as in Fig. 7, what is the time step?"

➔ Reply 29: Temporal resolution is 10 seconds. Info has been added to the figure caption.

"Anonymous Referee #2

Review of manuscript amt-2019-468
General aspects:

This is a well-written and interesting study showing how low cost metal oxide semiconductor
sensors (MOS) for methane (CH4) can be used to follow CH4 mixing ratios over
time in Greenland glacier ice caves. Results convincingly indicate that MOS sensors
can perform very well and this is promising for easier and less costly monitoring under
such conditions (very stable temperature and relative humidity). These tests are
important and I congratulate the authors for their careful and interesting work.
The authors are asked to consider the specific comments below in the revision of the
manuscript.

Specific comments (numbers refer to line numbers):
15. Please define CRDS in abstract. Some readers may not be familiar with cavity
ring-down spectrometry."

➔ Reply 30: Corrected.

"19-20: What was MBE selected instead of MAE or RMSE? With MBE, positive and
negative bias cancel out which is not desirable. Please consider using RMSE or MAE
instead."

➔ Reply 31: This could be an area prone to confusion, and we agree to resolved this in the revised
manuscript by including both the mean bias error (MBE), mean absolute error (MAE) and root mean
square error (RMSE) in table 1, and replace the MBE in the abstract with the RMSE value.

"97-98. Is it really correct that the conductivity increase with gas concentration as indicated
here? Does not the output voltage increase with CH4 mixing ratio due to increasing
resistance at higher CH4 levels, which would mean reduced conductivity?"

➔ Reply 32: In the description of the circuit (https://www.figaro.co.jp/en/product/docs/tgs2611-
e00_product%20information%28en%29_rev00.pdf), the output voltage of the sensor ($V_{RL}$) is measured
across a voltage divider with a fixed load resistor. The resistance of the variable resistor (Rs) (i.e. the
sensing element itself) can be calculated based on the measured output voltage ($V_{RL}$) according to the
following formula:

[Figure]

As the reviewer correctly points out, the output voltage ($V_{RL}$) increased with increasing CH4
concentration. In our setup, we used a load resistor of 10 kOhm and a circuit voltage ($V_C$) of 5 volt.

As argument for why we believe the description in the MS is correct consider the following simplified example: With an output value of say 1.5 volt (arbitrary lower CH4 concentration), the sensor resistance would be Rs: (5V/1.5V – 1) * 10kOhm = 23 kOhm, where a higher output value of say 2.5 volt (arbitrary higher CH4 concentration) would give an sensor resistance of 10 kOhm. In this way, higher CH4 concentrations produce lower sensor resistance or higher sensor conductivity.

"120. Eq.1: What is R0 in Figure 3? Is it equivalent to Rs? If so, please consider using consistent notation in both text, figures and tables."

➔ Reply 33: $R_0$ is equal to the sensor resistance $R_s$ under the defined zero gas conditions, to which all other measurements are normalized to (i.e. to calculate the $R_s/R_0$ ratio). Figure 3 only shows the simplified schematic of the electrical circuit, which does not include a notation of the calculated value of sensor resistance at zero gas conditions.

"139-148: Please here explain why the smoothing was needed. An explanation is given later in the text, but it would be good for understanding to provide the explanation here."

➔ Reply 34: We have revised the sentence to be: "In order to compensate for potential effects of micro-turbulent mixing of subglacial air with atmospheric air (see also section 3.3), the measured raw time series data from the MOS were smoothed using simple exponential smoothing according to Eq. (2):…….."

"155-160 and elsewhere. At less stable conditions than in the ice cave studied here, it would be challenging to have zero gas and sample gas with the same water concentrations. Hence, correction to humidity seems needed. Please see doi.org/10.5194/bg-2019-499 for detailed analyses of ways to correct for humidity and temperature to derive more generally applicable calibration curves."

➔ Reply 35: Thank you for providing the reference to an exciting and very relevant study, which has been published for discussion at a similar point in time as this study. We will include the reference to the list of references and include it in the revised discussion of the potential effect of water vapor and temperature on the sensor measurements.

"163-165- Unclear how the rather poor fit in Figure 6 between MOS and CRDS could be translated into the very close fit in Figure 7. Please clarify this in the manuscript."

➔ Reply 36: We are not sure we understand what is meant by poor fit in figure 6 ($R^2$ = 0.98 / p-value = 0.001). However, there is a notable spread in some of the values of under higher concentration levels under more turbulent environmental conditions at the measurement point. However, it is difficult to visualize all of the 37.140 data points in only single figure without a great number of similar data points are visually stacked on top of each other, thereby not being visible in the figure. In this way, the apparent higher degree of spread in the upper CH4 levels are a product of the turbulent data (which are non-dominating for the statistical model) having a visual prevalence over the non-turbulent data points which dominate the statistical model.

The following text has been added to the revised MS in section3.2. to clarify the issue:

*"A total of 37,140 data points are included in the regression model for converting the RS/RO\* ratios to CH4 concentrations. Inclusion of data points from the micro-turbulent periods produces a noisy visualization of the calibrations data at higher CH4 concentration levels (Fig. 6). However, this apparent noise is primarily a visual artefact that does not have significance for the underlying calibration statistics, which shows excellent statistical agreement between the independent and dependent variables (R2 = 0.98; p-value: 0.001)."*

"163-174. Could the deviation between the lab and the field be due to any other factors?"

➔ Reply 37: It is possible, yes, but further experimental and calibration work would be needed to bring more certainty to this issue.

"239-240. This statement gives the impression that the MOS are accurate to 10 ppb
level. Is this really correct? This is orders of magnitude better than others have found.
The mean bias error is risky to use because negative and positive errors cancel out.
Please consider using RMSE as indicator of MOS performance."

➔ Reply 38: We agree to use the RMSE as suggested and have updated the sentence accordingly.

"243-254. Would not field calibration also be an option as done here and suggested
in doi.org/10.5194/bg-2019-499? Given the low temperature - what was the absolute
humidity which is what influence sensors more than RH?"

➔ Reply 39: We agree. The original wording was chosen to reflect what has previously been done. The sentence is updated with the inclusion of the suggested reference. The saturated water vapor or absolute humidity in the thermally buffered measurement environment around 0 degrees C is approximately 5 g/m$^3$ (see http://hyperphysics.phy-astr.gsu.edu/hbase/Kinetic/watvap.html).

"305-307. Some of this is addressed in doi.org/10.5194/bg-2019-499 which could be
worth citing."

➔ Reply 40: We agree. Citations have been added.

"323-324. Please see previous comments regarding MBE vs RMSE."

➔ Reply 41: We have rephrased to focus on the RMSE

"484-485. Please clarify what FIgure 6 shows in relation to Figures 7 and 8. The offset
between the sensor and CRDS data are much greater in Figure 6 than in Figure 7 and
8. Figure 6 looks more like what could be expected from theses sensors, while the
fit versus the CDRS in Figure 7 and 8 is extremely close (looks fantastic and almost
too good to be true, and it is hard to understand how the calibration equations provided
could correct all the offset in Figure). Hence, clarifying the differences between Figure
6 vs 7 and 8 seem very important for fully understanding the study and proper sensor
use."

➔ Reply 42: Please see reply 36 for the issue of data fit (Figure 6) and reply 44 for precision issue. In short, our results are in line with the finding of also https://doi.org/10.5194/amt-2019-402, where the

authors conclude *"… that the TGS 2600 sensor can provide data of research-grade quality if it is adequately calibrated and placed in a suitable environment where cross-sensitivities to gases other than $CH_4$ is of no concern."*

"490-496. Legend of Figure 7 has many abbreviations. Please consider to define or spell them out to make it easier to understand the figure independently from the main text? Also it would be of great interest to readers to add humidity to the figure."

➔ Reply 43: Definitions have been added to captions for Figure 7.

With respect to the relative humidity. The resolution of the used RH sensor (S-THB-M008 from Onset;) is stated by the manufacturer to be 0.1 % RH (https://www.onsetcomp.com/products/sensors/s-thb-m008/). From our measurements, the results show a flat line of 100% relative humidity over the entire measurement period. We therefore chose to only describe this result by means of text (line 251 in original MS) and not show a flat in the figure.

[Figure]

To accommodate the comment from reviewer 2, we have expanded the description of the relative humidity measurements in the text, to better illustrate that RH was extremely stable in the measurement environment (resolution of sensor has been added in section 2.1 and text has been revised in section 3.4).

*"Measurements from the air-filled cavity under the GrIS document a very stable sampling environment with a relative humidity throughout the sampling period showing consistent readings of 100 % RH (data not shown) and only minor air temperature variations between approximately 0.05 °C during the night and 0.25 °C during mid-day (Fig. 7d)."*

**"Anonymous Referee #3**

In this paper, the authors studied the performance of a low-cost and low-power methane (CH4) sensing system prototype based on a metal oxide sensor (MOS) sensitive to CH4. The sensor was tested in a natural CH4 emitting environment at the Greenland Ice sheet (GrIS). The primary scientific importance of the study is that it provides a clear example on how the application of low cost technology can enhance our future understanding on the climatic feedbacks from the cryosphere to the atmosphere. The present study fits within the aim of this journal and the results are promising and interesting for future applications of low cost sensors.

The reviewer think that the paper can be published for open discussion and a main lack has been observed: - Low costs sensors from past studies show a 'drift' of the sensors response over the time. The authors do not cite this problem and neither they have tested it because a short experiment has been performed. This should be underline and future studies should include long term comparison between reference instrument and low cost sensor kit. The correction for the drift of the sensor will increase the final uncertainty related to the measurement and will also increase the cost of the field campaign because of the need of in situ continuous calibrations. The reviewer suggests to perform a study on the sensor drift over the months."

➔ Reply 44: We agree with #R3 that sensor drift may be an issue which potentially can limit the long-term applicability for the specific sensor, and that this issue should be addressed in our future research.

Another study has been published for discussion in AMT during the time period where our manuscript has been in review, namely [https://doi.org/10.5194/amt-2019-402] "*Long-term reliability of the Figaro TGS 2600 solid-state methane sensor under low Arctic conditions at Toolik lake, Alaska*" by Eugster et al., 2019. In this study, the authors evaluated the long-term stability of a similar $CH_4$ sensitive metal oxide sensor and conclude the following:

- Quotations from the abstract of amt-2019-402:

*"At weekly resolution the two sensors showed a downward drift of signal voltages indicating that after 10–13 years a TGS 2600 may have reached its end of life.…"*

*"Weekly median diel cycles tend to agree surprisingly well between the TGS 2600 and reference measurements during the snow-free season, but in winter the agreement is lower."*

*"We conclude that the TGS 2600 sensor can provide data of research-grade quality if it is adequately calibrated and placed in a suitable environment where cross-sensitivities to gases other than $CH_4$ is of no concern."*

- Quotation from amt-2019-402, ll. 208-210:

*"They [TGS 2600 sensors] provide encouraging results suggesting that with occasional (infrequent) calibration against a high-quality standard, e.g. using a traveling standard operating during a few good days with adequate coverage of the near-surface diel cycle of $CH_4$, TGS 2600 measurements might be suitable for the monitoring of $CH_4$ concentrations also in other areas."*

We have added this reference to the manuscript in section 3.5, where the issue of potential drift is discussed.